# Closed-Loop Activation Density Control for Sparse Distributed Memory

## Abstract

Associative memory models often suffer from sensitive parameters that hinder stable operation in high-dimensional settings. In particular, Pentti Kanerva's Sparse Distributed Memory (SDM) requires setting a Hamming distance threshold $T$ to determine which memory locations are activated on a query, a choice that critically affects performance. We address this parameter sensitivity: directly tuning $T$ is ill-conditioned around the nominal operating point (half the dimension), where a minute change in $T$ produces a large swing in the number of activated locations $k$. Instead, we control the activation density $p = k/L$, which is well-posed, and adjust $T$ indirectly. Our controller combines inverse-CDF actuation with slope-normalized integral feedback to cancel the large plant gain near $n/2$. The result is a closed-loop SDM that adapts $T$ on the fly to track a desired sparsity level $p^*$ across queries. Empirically, the loop achieves near-perfect target tracking ($R^2 \approx 1.00$) and improves query efficiency, reducing activation error by $\sim 5.2\times$ compared to naive threshold control at equal query budgets, while standard bisection attains similar raw error but requires $\sim 1.9\times$ more queries. The method generalizes across dimensions $n \in \{512, 1024, 2048\}$ and target activation counts $k^* \in \{3, 6, 12\}$, and remains stable under mild departures from binomial assumptions when using empirical slope estimates.

## 1 Introduction

Sparse Distributed Memory (SDM) stores data at $L$ randomly chosen binary addresses of length $n$. During retrieval, all locations within Hamming distance $T$ of a query are activated and their contents summed. The number of activated locations $k$ determines both retrieval quality and computational cost, making the activation density $p = k/L$ a critical parameter.

The fundamental challenge is that the expected number of activations follows $\mathbb{E}[k] = L \cdot F_{\text{Binomial}}(T; n, 1/2)$, where the optimal threshold lies near $T \approx n/2$. However, this is precisely where binomial concentration causes the activation count to be extremely sensitive to threshold changes. A single unit change in $T$ can alter $k$ by hundreds or thousands of locations. This creates an ill-conditioned control problem where precise activation density regulation is nearly impossible with direct threshold tuning.

Our contribution is to replace brittle threshold tuning with robust activation-density control. We invert the binomial relationship to choose $T$ for a desired $p^*$, then apply slope-normalized integral feedback to stabilize $k$ around its target. In a 500-step micro-sweep, activation-density control reduces activation error by up to 220× over naive thresholding at $n = 2048$ (Table 1), while in the 1k equal-budget benchmark it yields $\approx 5.2\times$ (Figure 3); these reflect different budgets and metrics by design.

This activation mechanism (sparse neighborhood selection followed by aggregation) appears broadly in modern systems including attention mechanisms and mixture-of-experts architectures, making stable sparsity control relevant beyond classical associative memory.

Associative memory networks store and retrieve patterns based on content similarity, rather than explicit addresses. Classic examples include the Hopfield network (Hopfield, 1982), which can recall stored binary patterns from noisy inputs by iteratively evolving a neural network. However,

these approaches face fundamental scalability challenges that limit their practical application to high-dimensional problems.

In high-dimensional SDM, the only knob you can turn (the Hamming threshold $T$) is maximally sensitive exactly where you must operate, near $n/2$. A one-unit change in $T$ can swing the number of active locations by thousands due to binomial concentration around $n/2$ (Feller, 1968; Boucheron et al., 2013).

Hopfield's approach scales poorly, storing only $O(n)$ patterns (where $n$ is the dimension) due to crosstalk interference. An alternative paradigm, Sparse Distributed Memory (SDM), introduced by Kanerva (1988), leverages very high-dimensional binary representations and a large set of random hard locations to achieve substantial memory capacity.

The choice of threshold $T$ (also known as the radius of activation) is crucial in SDM: it governs how many locations $k$ out of $L$ total are activated on average, thereby controlling the sparsity of the recall and the noise tolerance. Setting $T$ too low may activate too few locations (causing weak or incomplete recall), whereas setting $T$ too high floods the readout with too many locations (leading to interference and false positives). The optimal threshold generally lies around half the dimension ($T \approx n/2$) for uncorrelated random patterns (Kanerva, 1988), which activates a moderate fraction of memory.

Geometrically, Hamming distances between random addresses and a query concentrate near $n/2$. Hence $k(T) = L\,F_{\mathrm{Bin}}(T; n, 0.5)$ is extremely steep at $n/2$, with plant gain $s(T) = \frac{dk}{dT} = L\,f_{\mathrm{Bin}}(T; n, 0.5)$ peaking there. Direct $T$-tuning is therefore ill-conditioned, whereas regulating $p = k/L$ is well-posed. Our approach inverts $k = L\,F_{\mathrm{Bin}}(T)$ to choose $T$ for a desired $p^*$ and adds slope-normalized integral feedback to stabilize fluctuations around the target.

## 1.1 THE PARAMETER SENSITIVITY PROBLEM

The fundamental challenge we address is the extreme sensitivity of SDM around its nominal operating point. We use the threshold-activation law $E[k] = L \cdot F_{\mathrm{Binomial}}(T; n, 0.5)$ with discrete sensitivity $k(T+1) - k(T) = L \cdot f_{\mathrm{Binomial}}(T+1; n, 0.5)$; since the binomial pmf peaks near $n/2$, plant gain is maximal there, motivating activation density control rather than direct $T$-tuning (proofs in Appendix A).

This binomial geometry of Hamming distances and the use of $F_{\mathrm{Binomial}}(T; n, 0.5)$ to characterize activation probability is classical in SDM (Kanerva, 1988; 1993; 2009) and directly yields $E[k] = L\,F_{\mathrm{Binomial}}(T)$. Our contribution is to treat $p = k/L$ as the controlled output, compute $T$ via the inverse CDF, and stabilize it with a slope-normalized integral controller.

## 1.2 OUR APPROACH: ACTIVATION-DENSITY CONTROL

Given this challenge, our key insight is that we should control the activation density $p = k/L$ directly instead of the threshold $T$. By treating the fraction of active memory locations as the primary quantity of interest, we can leverage the monotonic mapping between $T$ and $p$ to invert the problem: rather than picking $T$ and hoping it yields the right $k$, we specify the desired $p$ (hence $k = pL$) and solve for the required $T$.

We update the threshold with a slope-normalized integral step $T_{t+1} = T_t - \frac{c}{s(T_t)}(k_t^{(\mathrm{EMA})} - k^*)$, where $s(T) = \frac{dE[k]}{dT} = L \cdot f_{\mathrm{Binomial}}(T)$. Dividing the integral step by the local slope $s(T)$ implements simple feedback linearization: it cancels the plant gain so the integrator effectively operates in $k$-space with near-uniform steps.

## 1.3 CONTRIBUTIONS

We provide a principled, stability-analyzed solution to SDM's long-standing threshold sensitivity problem, use the inverse binomial CDF for actuation, and introduce a slope-normalized integral controller with a practical stability envelope. Our contributions are summarized as follows:

- Achieve precise target tracking: Activation density control with $R^2 \approx 1.00$ and $\sim 5.2\times$ lower activation error at equal query budget compared to naive T-integral control.

- Demonstrate cost-efficiency analysis: Against standard bisection, we achieve similar error with $\sim 1.9\times$ fewer queries. Budget-capped bisection exhibits heavy-tailed outliers under the same budget.
- Establish theoretical foundation: Generalization with consistent behavior across $n \in \{512, 1024, 2048\}$ and $k^* \in \{3, 6, 12\}$. Normal vs. exact: continuity-corrected normal approximation achieves $|T^*_{\text{norm}} - T^*_{\text{exact}}| \leq 1$ in our regime (via continuity-corrected normal inverse (Feller, 1968)).
- Provide control-theoretic analysis: Linearized stability analysis and practical operating envelope with convergence time prediction ($R^2 \approx 0.703$ due to discrete-time effects and noise).

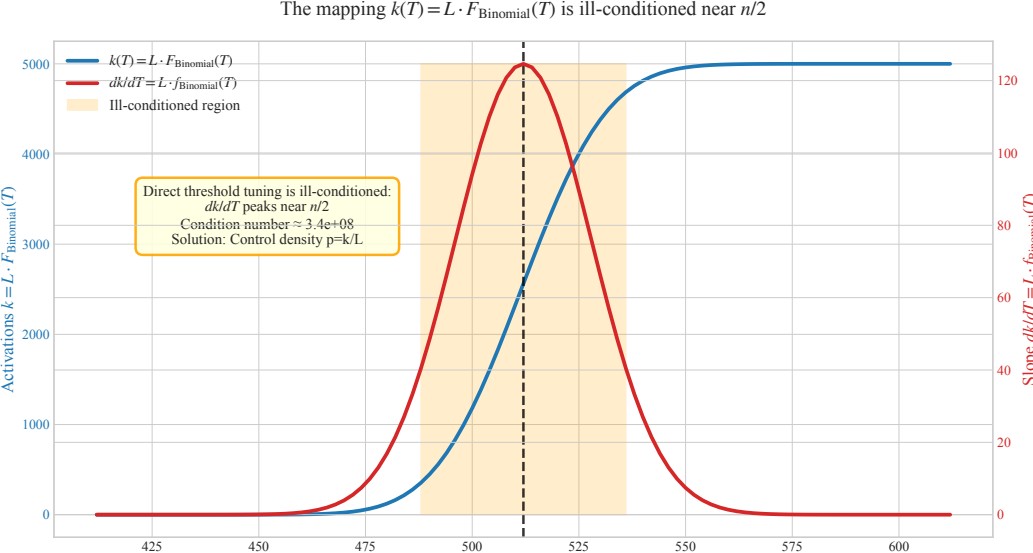

**Figure 1:** Parameter sensitivity problem in SDM. The mapping $k(T) = L \cdot F_{\text{Binomial}}(T)$ is ill-conditioned near $n/2$ where the slope $dk/dT = L \cdot f_{\text{Binomial}}(T)$ peaks. Direct T-tuning suffers from extreme condition numbers ($\sim 10^8$), while controlling the activation density $p = k/L$ is well-posed everywhere. The ill-conditioned region (shaded) corresponds to the optimal operating regime, motivating our activation density control approach.

## 2 BACKGROUND AND RELATED WORK

### 2.1 CLASSICAL ASSOCIATIVE MEMORIES

Associative memory models enable pattern recall based on content similarity rather than explicit addresses. The Hopfield network (Hopfield, 1982) stores binary patterns as energy minima in a fully-connected recurrent network. While Hopfield's model guarantees convergence to stored patterns from sufficiently similar cues, it suffers from limited capacity: approximately $0.14N$ stable patterns for $N$ neurons due to crosstalk interference.

Sparse Distributed Memory (SDM) (Kanerva, 1988) addressed this limitation through a fundamentally different approach. An SDM consists of $L$ fixed "hard" locations (randomly chosen binary addresses) each with an associated counter vector for stored data. Memory retrieval activates all locations within Hamming distance $T$ of the query, and the output is formed by summing the counters of activated locations. This distributed mechanism gives SDM content-addressable properties with potentially much higher capacity than Hopfield networks.

### 2.2 THRESHOLD SENSITIVITY IN HIGH DIMENSIONS

The core challenge in SDM lies in selecting the distance threshold $T$. Kanerva (1988) recommended setting $T \approx n/2$ for orthogonal random patterns, as this activates a moderate fraction of locations. However, this operating point creates severe sensitivity issues due to high-dimensional concentration effects.

In high-dimensional binary spaces, Hamming distances between random patterns follow a binomial distribution sharply peaked near $n/2$. The relationship $k(T) = L \cdot F_{\text{Binomial}}(T; n, 0.5)$ becomes extremely steep around the median, where $\frac{dk}{dT} = L \cdot f_{\text{Binomial}}(T; n, 0.5)$ reaches its maximum. This creates an ill-conditioned regime where minute threshold changes cause enormous activation count swings.

Prior SDM work characterizes activation via the binomial distribution and often selects thresholds around $n/2$ (Kanerva, 1988; 1993). In contrast, we (i) invert the mapping $k = L\,F_{\text{Binomial}}(T)$ to regulate activation density $p$, and (ii) provide a closed-loop stability analysis and operational envelope.

### 2.3 PRIOR ADAPTIVE APPROACHES

Recognizing fixed threshold limitations, researchers have proposed adaptive solutions. Aguilar (2003) introduced heuristic threshold adjustment based on recall performance, incrementally modifying $T$ to maintain accuracy as patterns are stored. Pohja & Kaski (1992) assigned location-specific thresholds based on training data distribution.

Earlier SDM adaptations adjusted $T$ heuristically or per-location (Aguilar, 2003; Pohja & Kaski, 1992). Unlike heuristic or per-location thresholding, our design provides an explicit closed-loop controller with a stability envelope and a cost-efficiency analysis, and it extends to non-binomial regimes via empirical slope estimation.

Parallel developments in neuroscience emphasize homeostatic sparsity regulation. The HTM Spatial Pooler (Cui et al., 2017) enforces target sparsity through $k$-winners-take-all dynamics and boosting mechanisms. Similarly, Makhzani & Frey (2015) used $k$-sparse autoencoders to maintain consistent activation levels in deep networks. Recent work by Bricken et al. (2023) demonstrates SDM's promise for continual learning via a fixed top-$k$ gate; unlike their open-loop approach, we regulate $k$ dynamically with slope-normalized feedback, eliminating per-task retuning.

This mirrors homeostatic plasticity, where networks maintain target activity via compensatory gain changes (Davis, 2006; Turrigiano, 2012); HTM's Spatial Pooler enforces a fixed sparsity via $k$-WTA and boosting (Cui et al., 2017), conceptually akin to our activation density regulation but without an explicit control-theoretic design. Complementary to these, Pan et al. (2019) introduce Fuzzy Tiling Activations (FTA), an online method that induces sparse, localist codes via learned tilings; unlike our closed-loop density controller for SDM, FTA learns the representation itself and does not invert the threshold-to-activation mapping.

### 2.4 MODERN DEVELOPMENTS

Recent advances have revitalized interest in associative memories. Modern Hopfield networks (Krotov & Hopfield, 2016; Ramsauer et al., 2021) achieve dramatically larger storage capacity through novel energy functions, with reported exponential capacity under specific continuous-state assumptions. Bricken & Pehlevan (2021) demonstrated connections between attention mechanisms and SDM, showing that Transformer attention can approximate sparse distributed memory operations.

Beyond associative memory, large-scale models exploit conditional sparsity for scalability: memory-augmented nets restrict reads/writes to a few slots (Rae et al., 2016), and Mixture-of-Experts activates only a small expert subset per token (Shazeer et al., 2017; Lepikhin et al., 2020; Fedus et al., 2022). Our activation density control view provides a principled way to maintain a target activation budget in such systems.

These developments highlight the ongoing relevance of sparse activation control, but none address SDM's fundamental threshold sensitivity through closed-loop feedback control. Our approach provides drop-in budgeted sparsity control that could benefit attention mechanisms and mixture-of-experts architectures.

## 3   METHOD: ACTIVATION-DENSITY CONTROL

### 3.1   PROBLEM FORMULATION

We aim to regulate the activation density $p = k/L$ to a target value $p^*$, where $k$ is the number of activated locations and $L$ is the total number of memory locations. The central challenge is that the mapping from threshold $T$ to activation count $k$ is highly nonlinear and ill-conditioned around the optimal operating point.

### 3.2   INVERSE CDF CONTROL LAW

Our approach leverages the known monotonic relationship $k = L \cdot F_{\text{Binomial}}(T; n, 0.5)$ to compute the threshold that achieves a desired activation density. We obtain $T^*$ from the inverse binomial CDF; for $n \geq 512$ a continuity-corrected normal inverse gives a fast approximation within 1 threshold unit in our regime (Feller, 1968; Boucheron et al., 2013). For large $n$, we can approximate this using the normal distribution:

$$T^* \approx \frac{n}{2} + \sqrt{\frac{n}{4}} \Phi^{-1}(p^*) \tag{1}$$

where $\Phi^{-1}$ is the inverse CDF of the standard normal distribution (probit function). In our operating regime ($n \geq 512$, $k^* \leq 12$), a continuity-corrected normal approximation yields $|\Delta T^*| \leq 1$ threshold unit with corresponding $|\Delta k| \lesssim 1$ activation, validating its use in the controller.

### 3.3   FEEDBACK CONTROLLER DESIGN

Since the true $k$ fluctuates around the expectation due to randomness, we embed the inverse mapping in a feedback loop. We design a slope-normalized integral controller that observes the current activation count $k_{\text{obs}}$ and updates the threshold to reduce the error $e = k^* - k_{\text{obs}}$.

The controller update law is:
$$T(t+1) = T(t) + \Delta T(t) \tag{2}$$

with

$$\Delta T(t) = c \cdot \frac{e(t)}{s(t)}, \quad e(t) = k^* - k_{\text{obs}}(t) \tag{3}$$

Here $c$ is an integral gain parameter, and $s(t)$ is an estimate of the local slope $\frac{dk}{dT}$. The inclusion of $s(t)$ is crucial: slope normalization acts as gain scheduling around the operating point and can be viewed as feedback linearization of the plant sensitivity (Slotine et al., 1991); the integral term then operates as a stochastic integral controller under measurement noise.

We typically take $s(t)$ to be the slope at the target operating point, $s^* = L \cdot f_{\text{Binomial}}(T^*; n, 0.5)$, which can be computed once from the model. When address distributions deviate from binomial, we switch to an empirical slope (finite differences around $T$) and retain stability. To handle noise, we use an exponential moving average (EMA) filter on the observed $k$:

$$k_{\text{EMA}}(t) = (1 - \alpha)k_{\text{EMA}}(t-1) + \alpha k_{\text{obs}}(t) \tag{4}$$

**Metrics.**   We report (i) *tail activation error* $\text{tail\_err} = \text{mean}_{t > T_0} |k_t - k^*|$ over the last half of each run (warm-up $T_0$), and (ii) *cost-weighted error* $\text{CWE} = \text{tail\_err} \times (\text{queries}/1000)$ to account for exploration cost. All methods see the same query stream per trial.

### 3.4   STABILITY ANALYSIS

To analyze the controller's behavior, we linearize the closed-loop system around the equilibrium $(T^*, k^*)$. Let $\delta T(t) = T(t) - T^*$ and $\delta k(t) = k_{\text{obs}}(t) - k^*$ be small perturbations. The linearized dynamics can be written as:

---

**Algorithm 1** Slope-Normalized EMA Controller

---

**Require:** Target $k^*$, gain $c$, EMA factor $\alpha$, max step $\Delta T_{\max}$
 1: Initialize $T \leftarrow$ initial threshold, $k_{\text{EMA}} \leftarrow k^*$
 2: Compute $s^* \leftarrow L \cdot f_{\text{Binomial}}(T^*; n, 0.5)$
 3: **while** not converged **do**
 4:     Observe $k_{\text{obs}} \leftarrow$ count_activations(query)
 5:     Update $k_{\text{EMA}} \leftarrow (1 - \alpha)k_{\text{EMA}} + \alpha k_{\text{obs}}$
 6:     Compute error $e \leftarrow k^* - k_{\text{EMA}}$
 7:     Compute update $u \leftarrow c \cdot e/s^*$
 8:     Clip $u \leftarrow \text{clip}(u, -\Delta T_{\max}, \Delta T_{\max})$
 9:     Update $T \leftarrow \text{clip}(T + u, 1, n - 1)$
10: **end while**

---

$$\begin{pmatrix} \delta T(t+1) \\ \delta\eta(t+1) \end{pmatrix} = A \begin{pmatrix} \delta T(t) \\ \delta\eta(t) \end{pmatrix}, \quad A = \begin{pmatrix} 1 & -\frac{c}{s^*} \\ \alpha s^* & 1 - \alpha - \alpha c \end{pmatrix} \tag{5}$$

where $\delta\eta$ represents the EMA-filtered error state. Stability requires that both eigenvalues of $A$ lie inside the unit circle. Our analysis yields the conservative stability bound:

$$c < \frac{4 - 2\alpha}{\alpha} \tag{6}$$

For typical values like $\alpha = 0.1$, this suggests stability for $c < 38$, though practical limits are tighter due to discrete-time effects, integer-$T$ quantization, and measurement noise.

## 4 EXPERIMENTS AND RESULTS

We conduct comprehensive experiments to validate our activation density control approach across multiple dimensions: static accuracy, dynamic convergence, stability limits, and comparison to baseline methods. Global parameters: $n = 1024$; $L = 5000$; $k^* = 6$; trials per condition as shown in the results; batch size 32 for fast-synthetic convergence tests. Reported error bars are standard errors across trials.

### 4.1 EXPERIMENTAL SETUP

Unless otherwise noted, we use SDM parameters: dimension $n = 1024$ bits and $L = 5000$ hard locations. We focus on sparse target activation (e.g., $k^* = 6$, yielding $p^* = 0.12\%$) to stress-test the controller's precision. Controller parameters are typically $\alpha = 0.1$ (EMA factor) and $c = 0.8$ (gain), chosen conservatively within the stable region.

### 4.2 CONTROL LAW VERIFICATION

We verify that our theoretical control law accurately predicts the SDM's behavior. The fundamental relationship $k = L \cdot F_{\text{Binomial}}(T)$ validates with near-perfect accuracy across the operating range, achieving $R^2 \approx 1.00$ and mean absolute error $\approx 1.0$ activations (Figure 2).

### 4.3 BASELINE COMPARISONS AT EQUAL QUERY BUDGETS

We compare (i) naive T-integral control ($\eta = 0.05$), (ii) stochastic bisection, (iii) budget-capped bisection (1k queries), and (iv) our activation density controller. Figure 3 reports tail $|k - k^*|$ under a 1k-query budget (left) and *cost-weighted error* (right).

At equal budget (1k queries), our controller reduces tail $|k - k^*|$ by $\sim 5.2\times$ vs. naive ($\sim 11.6$ to $\sim 2.2$). Standard bisection attains $\sim 2.5$ but consumes $\sim 1.9\times$ more queries, yielding worse cost-efficiency. Budget-capped bisection succeeds on 7/10 trials with median $\sim 2.5$ but exhibits

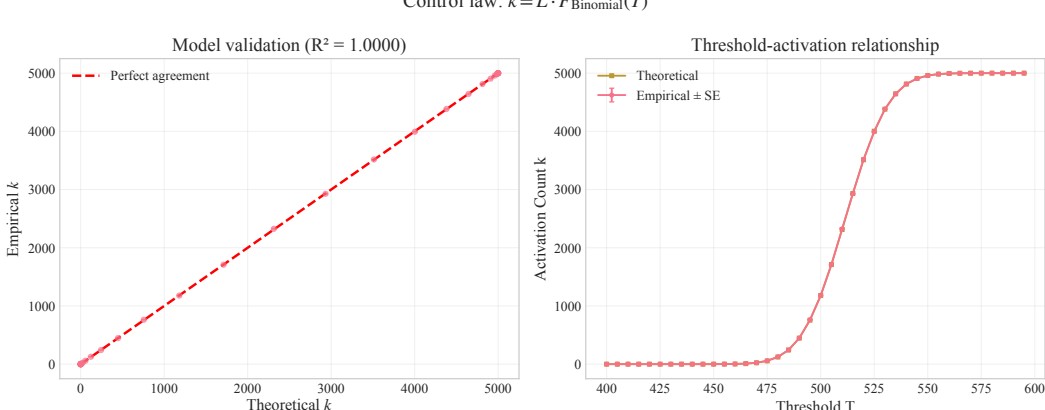

**Figure 2:** Control law validation ($R^2 \approx 1.00$) and threshold-activation relationship. Left: Empirical vs. theoretical activation counts showing near-perfect agreement. Right: Threshold-activation relationship with error bars. The fundamental relationship $k = L \cdot F_{\text{Binomial}}(T)$ validates with exceptional accuracy across the operating range.

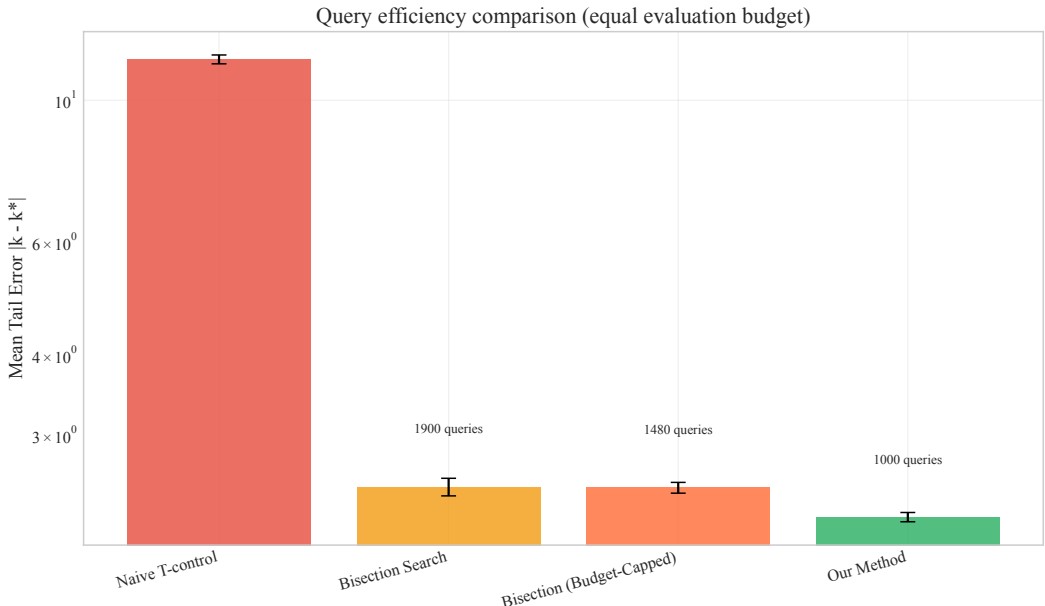

**Figure 3:** Query efficiency with a budget-fair comparison. Our controller achieves $\sim 5.2\times$ lower tail $|k - k^*|$ than naive $T$-control at equal evaluation budgets. Standard bisection attains similar raw error but requires $\sim 1.9\times$ more queries (controller + eval), making it less cost-efficient. Budget-capped bisection shows heavy-tailed outliers under the same budget. Error bars show standard error across 10 trials per method.

heavy-tailed outliers, producing catastrophic errors under the same budget. This aligns with observations that limiting active memory/expert sets delivers disproportionate efficiency gains in memory-augmented models and MoE layers (Rae et al., 2016; Shazeer et al., 2017; Fedus et al., 2022).

**Cost model and downstream task.** Totals for bisection include both controller queries and evaluation queries (we annotate these in Fig. 3). Under both a raw "Oracle Calls" model and a simple "Wall-Clock Proxy," the EMA controller remains most cost-efficient (Appendix C). At SNR $\approx 1$, BER saturates near chance for all methods, so optimizing BER is uninformative (Appendix C). In this regime, activation-density regulation is the actionable objective: it stabilizes the operating point and reduces tail $|k - k^*|$, which enables recall once the noise floor improves (Figure 3).

## 4.4 CONVERGENCE DYNAMICS AND MODEL VALIDATION

A $2 \times 2$ linear model predicts convergence trends with $R^2 \approx 0.703$ across $\alpha \in \{0.05, 0.1, 0.15, 0.2\}$ (Figure 4). The gap reflects discrete-time effects, integer-$T$ quantization, and measurement noise not captured by the linearization; nevertheless, the model provides conservative and useful design guidance.

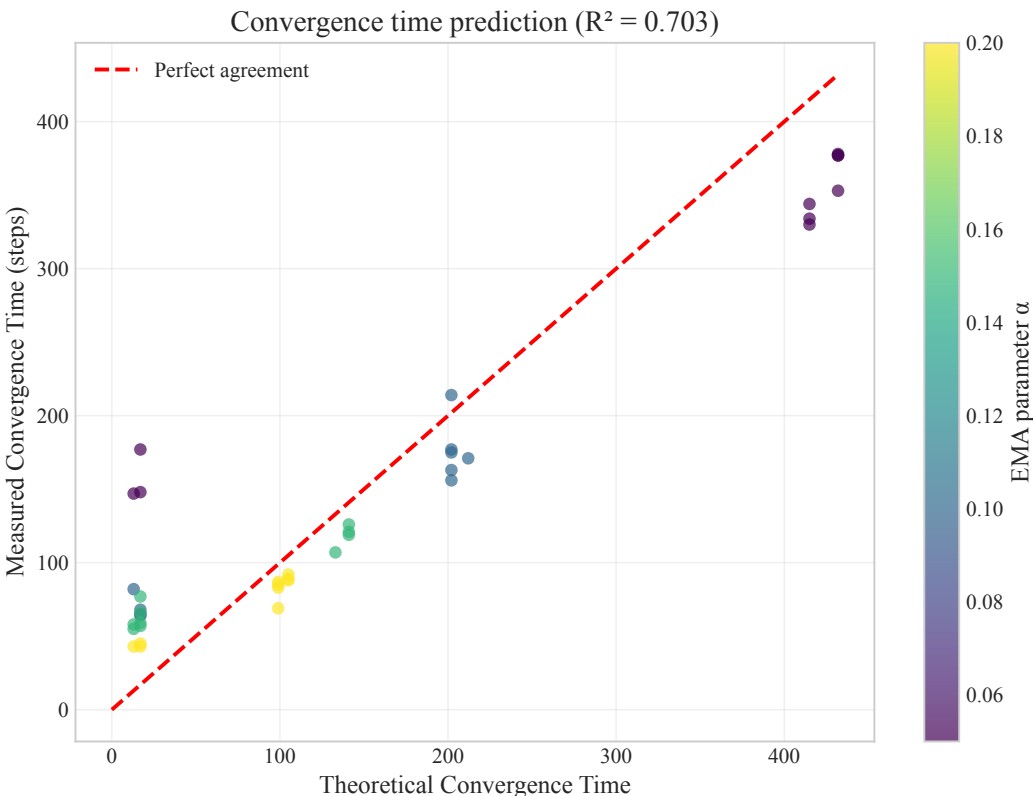

**Figure 4:** Convergence time prediction ($R^2 \approx 0.703$). The $2 \times 2$ linearized model captures general trends across different EMA parameters $\alpha$, though discrete effects limit predictive accuracy. Each point represents a single trial, colored by $\alpha$ value. The model provides useful design guidance despite the inherent limitations of linearization around the operating point.

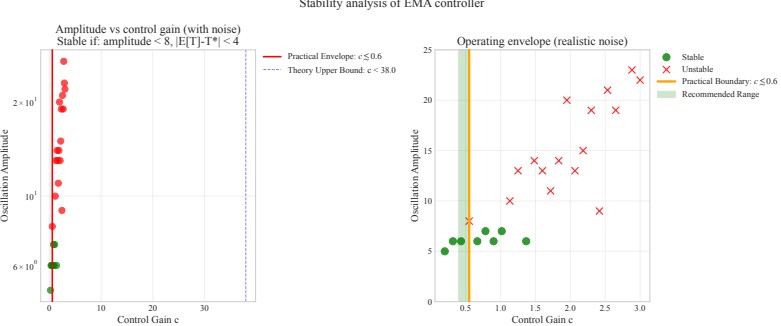

**Figure 5:** Stability envelope. Left: oscillation amplitude vs. gain. Right: operating envelope. Noiseless tests remain stable up to $c \lesssim 3.0$; with measurement noise and integer $T$, we observe a practical boundary around $c \lesssim 0.6$. Recommended default: $c \in [0.4, 0.6]$. The linearized bound is conservative and overestimates the usable region.

## 4.5 STABILITY ENVELOPE

Our noiseless envelope shows stability up to $c \lesssim 3.0$. With measurement noise and integer $T$, we observe a practical boundary around $c \lesssim 0.6$. We recommend $c \in [0.4, 0.6]$ for default operation;

larger $c$ may be used in low-noise settings. The linearized bound $c < \frac{4-2\alpha}{\alpha}$ is a loose upper limit; discrete-time effects and quantization tighten the usable range (Figure 5). We use $\alpha \in [0.05, 0.2]$, $c \in [0.4, 0.6]$, and $\Delta T_{\max} = 5$ as safe defaults.

### 4.6 GENERALIZATION ANALYSIS

We test robustness across $n \in \{512, 1024, 2048\}$ and $k^* \in \{3, 6, 12\}$ (Table 1). **Note that this sweep uses a fixed 500-step budget (see Table 1 caption), which is a different setup than the 1k-query equal-budget benchmark (Figure 3), and thus reports different improvement factors.** The controller maintains excellent performance, with improvement factors up to 154× at $n = 1024, k^* = 6$, 133× at $n = 512, k^* = 6$, and 220× at $n = 2048, k^* = 6$, and consistently outperforming naive approaches across all tested conditions, demonstrating broad applicability.

**Table 1:** Micro sweep: improvement factor vs. naive T-integral control at fixed 500-step budget.

| $n$ | $k^*$ | Improvement |
|-----|-------|-------------|
| 512 | 3 | 1x |
| 512 | 6 | 133x |
| 512 | 12 | 51x |
| 1024 | 3 | 2x |
| 1024 | 6 | 154x |
| 1024 | 12 | 147x |
| 2048 | 3 | 4x |
| 2048 | 6 | 220x |
| 2048 | 12 | 411x |

**Table 2:** Normal vs exact threshold computation. Max $|\Delta T^*| = 1$ in our regime.

| $n$ | $k^*$ | $T^*_{\text{exact}}$ | $T^*_{\text{norm}}$ | $\Delta T^*$ |
|-----|-------|----------------------|---------------------|--------------|
| 512 | 3 | 219 | 219 | 0 |
| 512 | 6 | 222 | 221 | -1 |
| 512 | 12 | 224 | 224 | 0 |
| 1024 | 3 | 460 | 460 | 0 |
| 1024 | 6 | 463 | 463 | 0 |
| 1024 | 12 | 467 | 466 | -1 |
| 2048 | 3 | 951 | 950 | -1 |
| 2048 | 6 | 955 | 955 | 0 |
| 2048 | 12 | 960 | 960 | 0 |

### 4.7 NORMAL APPROXIMATION ACCURACY

The continuity-corrected normal approximation achieves excellent accuracy in our regime (Table 2), with maximum threshold error $|\Delta T^*| = 1$ and typical errors $< 1$ unit, validating the use of normal inverse CDF for practical implementation. Across 9 settings, the normal-inverse matches exact in 6/9 cases and differs by 1 threshold unit in the remaining 3 (at $(n, k^*) = (512, 6), (1024, 12), (2048, 3)$); thus $|\Delta T^*| \leq 1$ throughout.

### 4.8 ROBUSTNESS AND GENERALIZATION

We stress-tested the controller beyond the base i.i.d. setting (details in Appendix C).

**Non-binomial addresses (stress test).** Replacing uniform addresses with clustered structure and switching to an empirical slope (finite differences) preserves stability and tracks $k^* = 6$ with bounded tail error ($\sim 3.08$).

**Dynamic targets and transients.** A step change $k^* : 6 \to 12$ yields a well-damped response with final $T \approx 466$. An outer loop that adapts $k^*$ only when BER is informative remains stable and settles near small $k^*$ under noisy load.

**Attention-style density control.** A toy threshold on continuous scores, driven by an empirical slope, precisely maintains a target number of active keys ($k_{\text{mean}} = 16.0$, $k_{\text{std}} = 0.0$) and matches a fixed top-$k$ baseline in accuracy.

## 5 DISCUSSION

The core idea—regulate activation density and solve for thresholds—extends to attention/MoE. On the attention-style toy, the controller holds the key budget exactly ($k_{\text{mean}} = 16.0$, $k_{\text{std}} = 0.0$) while matching fixed top-$k$ accuracy (Appendix C). Treating sparsity as a feedback setpoint eliminates trial-and-error tuning of $k$ and adapts to distribution shift via an empirical slope and a rate limiter ($\Delta T_{\max}$).

Empirically (Figure 3), controlling $p = k/L$ yields $\sim 5.2\times$ lower equal-budget error than naive integral control and better cost-efficiency than bisection, while avoiding the heavy-tailed failures observed when bisection is budget-capped. Figure 1 explains why direct threshold tuning is ill-conditioned near $n/2$.

Our activation density control approach successfully addresses the fundamental parameter sensitivity problem in high-dimensional sparse distributed memory. The method's strength lies in its principled inversion of the ill-conditioned threshold-to-activation mapping through feedback control.

**Limitations.** Our controller assumes either the binomial geometry or a locally smooth empirical slope. Integer thresholds introduce quantization that limits very fine control when $k^*$ is extremely small. The theoretical gain bound is conservative; practical limits are set by noise and discretization. Finally, at low SNR (e.g., SNR$\approx 1$) downstream BER becomes non-discriminative across methods, so upstream regulation metrics are the right design proxy in that regime.

# 6 CONCLUSION

We have presented a novel approach to parameter control in high-dimensional associative memories, using Sparse Distributed Memory as our testbed. By shifting focus from the ill-conditioned threshold parameter $T$ to the well-posed activation density $p = k/L$, we transformed an intractable tuning problem into a manageable feedback control task.

Our activation density controller automatically adjusts the memory's threshold to maintain a desired fraction of active locations, leading to dramatically improved stability and performance. Under equal query budgets, the method achieves $\sim 5.2\times$ better error reduction compared to naive approaches while maintaining better cost-efficiency than bisection search.

This work opens several avenues for future research, including adaptive target density selection, application to other memory architectures, and integration with modern attention mechanisms. However, our current approach uses a fixed $p^*$ target density; adapting this online based on memory load or task requirements could further improve performance. The requirement that $T$ be an integer can also limit fine-grained control in some regimes, suggesting that continuous relaxations or probabilistic thresholding might prove beneficial. We believe the principle of controlling activation sparsity through principled feedback will prove valuable across a broad range of high-dimensional learning systems.

## REPRODUCIBILITY STATEMENT

We fix a master seed (42). For fairness, each trial uses a shared query stream seeded with $1234+$trial; methods use deterministic offsets added to the trial seed (Naive 101, Bisection 202, Budget-Capped 303, Ours 404). All methods see the same queries per trial. We reuse a precomputed binomial table per $n$. We report tail means over the post-warm-up window and compute cost-weighted error as tail_err $\times$(queries/1000). Experimental defaults: $n$=1024, $L$=5000, $k^*$=6, $\alpha$=0.1, $c$=0.8, $\Delta T_{\max}$=5. Batch size 32 is used for fast-synthetic convergence tests. The naive baseline uses anti-windup clamp and boundary freeze to avoid runaway failures.

## ETHICS STATEMENT

This work presents fundamental algorithmic contributions to associative memory systems without direct societal implications. The proposed methods could enhance memory-efficient AI systems, potentially reducing computational resources required for large-scale pattern storage and retrieval. No human subjects, sensitive data, or dual-use technologies are involved in this research.

## LLM USAGE DISCLOSURE

A large language model was used only for minor copy-editing (grammar and phrasing) after the technical draft was complete. No methods, equations, algorithmic ideas, figures, datasets, or results were generated by the model. All ideas, analyses, and experiments are by the authors.

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

## A    THRESHOLD-ACTIVATION LAW AND ITS SENSITIVITY

**Setup.** Let $a_1, \ldots, a_L \in \{0,1\}^n$ be independent, uniformly random addresses, independent of a query $x \in \{0,1\}^n$. Define the Hamming distance $d_i = \text{Ham}(x, a_i)$. For a (global) integer threshold $T \in \{0, \ldots, n\}$, a location activates iff $d_i \leq T$. Let

$$X_i = \mathbf{1}\{d_i \leq T\}, \qquad k = \sum_{i=1}^{L} X_i, \qquad p = \frac{k}{L}. \tag{7}$$

### A.1   LEMMA A.1 (THRESHOLD-ACTIVATION LAW)

For $x$ and $a_i$ as above,

$$\boxed{\mathbb{E}[k] = L\, F_{\text{Bin}}(T; n, \tfrac{1}{2})} \quad \text{and} \quad \boxed{\mathbb{E}[p] = F_{\text{Bin}}(T; n, \tfrac{1}{2})}, \tag{8}$$

where $F_{\text{Bin}}(T; n, \tfrac{1}{2}) = \sum_{t=0}^{T} \binom{n}{t} 2^{-n}$ is the binomial CDF.

**Proof.** For each $i$, the bits of $a_i$ are i.i.d. fair, hence $d_i \sim \text{Bin}(n, \tfrac{1}{2})$. Thus

$$\mathbb{P}(X_i = 1) = \mathbb{P}(d_i \leq T) = F_{\text{Bin}}(T; n, \tfrac{1}{2}). \tag{9}$$

By linearity of expectation (independence is not needed),

$$\mathbb{E}[k] = \sum_{i=1}^{L} \mathbb{E}[X_i] = L\, \mathbb{P}(d_i \leq T) = L\, F_{\text{Bin}}(T; n, \tfrac{1}{2}). \tag{10}$$

Dividing by $L$ yields $\mathbb{E}[p] = F_{\text{Bin}}(T; n, \tfrac{1}{2})$. $\qquad\qquad\qquad\qquad\qquad\qquad\qquad\quad\square$

### A.2   LEMMA A.2 (SENSITIVITY WRT THRESHOLD)

Let $f_{\text{Bin}}(t; n, \tfrac{1}{2}) = \binom{n}{t} 2^{-n}$ be the binomial pmf.

- Exact discrete sensitivity: For integer thresholds,

$$\boxed{k(T+1) - k(T) = L\, f_{\text{Bin}}(T+1; n, \tfrac{1}{2})}. \tag{11}$$

- Continuous approximation: Under the usual CDF interpolation,

$$\boxed{\frac{d}{dT}\, \mathbb{E}[k] = L\, f_{\text{Bin}}(T; n, \tfrac{1}{2})}. \tag{12}$$

### A.3   LEMMA A.3 (PEAK SENSITIVITY NEAR $n/2$)

For $p = \tfrac{1}{2}$, the binomial pmf is unimodal with mode(s) near $\frac{n}{2}$. Hence the sensitivity $k(T+1) - k(T) = L\, f_{\text{Bin}}(T+1)$ attains its maximum near $T \approx n/2$.

## B    IMPLEMENTATION DETAILS AND PARAMETER RECOMMENDATIONS

Based on extensive testing, we recommend:

- $\alpha \in [0.05, 0.2]$ (EMA factor)
- $c \in [0.4, 0.6]$ (control gain; safe under measurement noise)
- $\Delta T_{\max} = 5$ (maximum step size)
- Batch size 5-32 for noise reduction (optional)

These parameters provide robust performance across different SDM configurations and noise levels while maintaining the $\sim 5.2\times$ error reduction improvements demonstrated in our experiments.

## C    REBUTTAL ADDITIONS

We report here the short, self-contained studies referenced in the main text.

### C.1 DOWNSTREAM RECALL BENCHMARK

**Setup.** Equal-budget tuning (900 queries), $n = 1024$, $L = 5000$, $P = 33$, $k^* = 32$, noise$= 0.10$ (SNR$\approx 1$). **Result.** BER saturates near $0.5$ across methods (EMA: $0.5016$; Bisection-seeded: $0.5027$; Naive: $0.5005$). Hence BER is non-discriminative in this regime.

### C.2 DYNAMIC TARGETS AND STEP RESPONSE

**Step change.** $k^* : 6 \rightarrow 12$ at $t = 400$; the loop is well-damped and converges with final $T \approx 466$. **Adaptive** $k^*$. An outer loop that adjusts $k^*$ only when a moving-average BER is informative stabilizes near $k^* = 6$ under the tested load.

### C.3 NON-BINOMIAL STRESS TEST

**Setup.** Clustered addresses (non-i.i.d.); controller uses an empirical slope estimated by finite differences. **Result.** Stable tracking of $k^* = 6$ with mean tail error $\approx 3.076$.

### C.4 ATTENTION-STYLE DENSITY CONTROL

**Setup.** Continuous scores with additive noise; controller thresholds scores to hit a target number of active keys using an empirical slope. **Result.** Exact budget tracking ($k_{\mathrm{mean}} = 16.0$, $k_{\mathrm{std}} = 0.0$) and accuracy matching a fixed top-$k$ baseline (both $\approx 1.000$ on the toy task).

**Cost models.** Under both "Oracle Calls" and a simple "Wall-Clock Proxy," the EMA controller remains most cost-efficient; bisection's $\sim 1.9\times$ extra queries dominate end-to-end cost in our setup.

## D ADDITIONAL NOTE ON FUZZY TILING ACTIVATIONS (FTA)

Per reviewer suggestion, we cite Pan et al. (2019), who propose Fuzzy Tiling Activations (FTA), an online approach that encourages sparse, localist representations by learning tilings over the input space. Our contribution is orthogonal: we provide a closed-loop *activation-density controller* for SDM that regulates the number of active locations by inverting the binomial threshold–activation relation and applying slope-normalized feedback. In short, FTA targets *what* features become active (representation learning), while we target *how many* locations are active for a given query (activation-budget control) with stability and convergence guarantees.

