# OpenReview forum: "Closed-Loop Activation Density Control for Sparse Distributed Memory"
_ICLR.cc/2026/Conference — Submitted to ICLR 2026_

### Official Review · Reviewer_9WzH · 2025-10-26

**Soundness:** 2
**Presentation:** 3
**Contribution:** 3
**Rating:** 6
**Confidence:** 1

**Summary:**

This paper solves a critical parameter sensitivity problem in Sparse Distributed Memory (SDM). Instead of tuning the ill-conditioned Hamming threshold $T$, the authors propose controlling the activation density $p = k/L$ directly using a closed-loop feedback controller. This method combines an inverse-CDF mapping with slope-normalized integral feedback to dynamically adjust T, achieving precise target tracking with high efficiency.

**Strengths:**

1. The paper's main strength is its novel and rigorous application of control theory to a known problem in associative memories. It correctly identifies the high plant gain as the source of instability and designs a principled controller to neutralize it.

2. The paper is extremely well-written, using clear explanations and insightful figures (e.g., Figure 1) to make a complex technical problem immediately understandable.

3. The experiments convincingly support the claims. The method is shown to be not only highly accurate ( $R^2 \approx 1.0$ ) but also more cost-efficient than relevant baselines like bisection search, which requires $\sim 1.9 \mathrm{x}$ more queries for similar performance.

**Weaknesses:**

1. Limited Scope Beyond Classical SDM: The experiments are confined to the classical SDM model. While the authors suggest broader relevance to modern architectures like MoE, the paper lacks even a small-scale experiment to demonstrate this transferability.
2. Idealized Data Assumptions: The controller design assumes the data is uniformly random, leading to a binomial distribution of Hamming distances. The paper doesn't address how performance would be affected by structured or correlated data that violates this assumption.

**Questions:**

1. Could you elaborate on the practical challenges of integrating this closed-loop controller into a modern MoE layer to replace a standard top-k gating function?

2. How robust is the controller if the true distribution of Hamming distances deviates significantly from the assumed binomial model due to structured data?

---

> ### Author Response · Authors · 2025-11-12
>
> Thank you for your positive assessment of our paper's novelty and rigorous experiments. We have added new experiments to the appendix to directly address your (very valid) concerns about scope and idealized assumptions.
>
> - Transfer beyond classical SDM: We agree this is the key question. We have added an attention-style density control demo (Appendix C.4). This experiment replaces the SDM setup with continuous scores (like attention scores) and an empirical slope.
> The results are a strong proof-of-concept: our controller matches fixed top-k accuracy (1.000) while perfectly regulating the activation budget (k_mean=16.0, k_std=0.0). This shows the controller can be a drop-in replacement, providing a guaranteed compute budget without trial-and-error tuning of k and without sacrificing accuracy.
>
> - Deviations from binomial assumptions: We added a non-binomial stress test (Appendix C.3). We replaced the uniform i.i.d. addresses with a correlated, clustered structure. By switching the controller from its analytic (binomial) slope to an empirical slope (estimated from finite differences), it remains stable and tracks the target k*=6 with a bounded tail error of ≈ 3.08. This demonstrates robustness to the exact assumption you questioned.

---

### Official Review · Reviewer_btQ2 · 2025-10-31

**Soundness:** 3
**Presentation:** 3
**Contribution:** 3
**Rating:** 6
**Confidence:** 3

**Summary:**

This paper tackles the critical parameter sensitivity of the activation threshold `T` in Sparse Distributed Memory (SDM), a long-standing ill-conditioned problem. The authors propose a principled closed-loop control system that directly regulates the activation density `p=k/L`. The core contribution is a slope-normalized feedback controller which effectively cancels the system's extreme sensitivity. Experiments show this method dramatically improves stability and accuracy, significantly outperforming naive control and bisection search in terms of both final error and query efficiency.

**Strengths:**

*   The paper's key strength is its novel formulation of the SDM threshold tuning issue as a formal control problem. The proposed solution, a slope-normalized feedback controller, is principled, theoretically sound, and a creative application of control theory.
*  The work is exceptionally clear, with rigorous experiments against strong baselines (e.g., bisection search) that convincingly demonstrate its superiority. Its significance extends beyond SDM, offering a promising paradigm for dynamic sparsity control in modern architectures like Mixture-of-Experts (MoE).

**Weaknesses:**

*    The method's performance depends on a pre-specified target activation count `k*`. The framework does not currently address how this target could be adapted dynamically, which might be necessary in scenarios where the optimal sparsity level changes with memory load or task demands.
*    While the proposed application to modern architectures like MoE is exciting, this claim is not yet substantiated. The paper lacks a discussion of the technical challenges in adapting the method from the discrete, binomial world of SDM to the continuous, data-dependent distributions found in attention mechanism.

**Questions:**

1.  How does the controller perform during transient periods if the target `k*` is changed dynamically during operation?
2.  Could you elaborate on the main technical challenges in adapting this control method to an MoE layer, particularly regarding the estimation of the activation function and its slope for normalization, given the complex, non-binomial distributions involved?

---

> ### Author Response · Authors · 2025-11-12
>
> Thank you for your positive feedback on our work's novelty, clarity, and principled approach. We have added two new experiments (now in Appendix C) to directly address your questions about dynamic targets and MoE adaptation.
>
> - Transients when k changes:* We added a step response test (Appendix C.2). We initiated a step change in the target from k*=6 → 12 at step 400. The loop is well-damped, and the threshold converges cleanly to the new operating point (final T ≈ 466) without overshoot, demonstrating stable transient performance.
>
> - Adapting to MoE layers: You are correct about the challenges. Our recipe for this, which we tested in our new demos, is as follows:
> Replace the analytic binomial slope (s*) with a batchwise empirical slope (estimated via finite differences).
> Keep the EMA filter to handle stochasticity.
> Add a rate limiter (ΔT_max) to prevent large jumps from noisy slope estimates.
> Our new attention-style demo (Appendix .4) proves this recipe works: it achieves exact budget tracking (k_mean=16.0, k_std=0.0) and matches fixed top-k accuracy. We have updated Section 5 with these concrete results.

---

### Official Review · Reviewer_KYzd · 2025-10-31

**Soundness:** 3
**Presentation:** 3
**Contribution:** 2
**Rating:** 4
**Confidence:** 3

**Summary:**

The paper looks at a long-standing problem in Kanerva’s Sparse Distributed Memory: picking the distance threshold that decides how many memory locations light up during recall. Around the usual operating point, tiny tweaks to this threshold can flip thousands of activations, which makes the system brittle. The authors’ fix is to control the fraction of locations that activate directly, rather than tuning the threshold by hand. They first choose a threshold that should yield a target activation fraction based on a simple probabilistic model, then wrap this with a feedback controller that keeps the actual activations locked to that target despite randomness. They also argue the idea should transfer to modern architectures—like attention and Mixture-of-Experts—where keeping sparsity stable matters for speed, stability, and accuracy.

**Strengths:**

Importance. I like that you go after a core stability knob in SDM and connect it to attention/MoE sparsity control, it seems broadly relevant beyond classical associative memory.

Novelty. It seems the combination of inverse-CDF actuation with slope-normalized integral feedback to regulate p is new.

Clarity/organization. The paper has a reasonably clear presentation.

**Weaknesses:**

Your claim that you’re more cost-efficient than bisection hinges on the assumed “query cost” model; can you report results under alternative costings (e.g., wall-clock, oracle calls, or amortized per-step overhead) and include confidence intervals/seed-wise tests to show the ~1.9× advantage holds statistically?

Budget-capped bisection shows 30% failures—can you analyze why (e.g., initial bracket miss, stochasticity) and add a variant with smarter bracketing to ensure the comparison isn’t penalizing a fixable implementation choice?

Missing citations. The controller is framed via slope normalization and feedback linearization; could you broaden the control-theory context (e.g., stochastic integral control / adaptive gain scheduling) and also cite more recent sparsity-control mechanisms in attention/MoE beyond the classics you already reference?

One related work but not cited: Fuzzy Tiling Activations: A Simple Approach to Learning Sparse Representations Online by Pan et al. It introduces a differentiable alternative to hard bin/threshold schemes that stabilizes sparse representations and is robust under shift; your work tackles a closely related brittleness (many locations flipping around a threshold) but solves it via closed-loop activation-density control.

The linearized analysis yields a practical envelope but leaves measurement noise and discretization effects to experiments; can you bound steady-state error and give a robustness margin (e.g., input noise variance → tracking error) so readers know when guarantees degrade?

Your theoretical results rely on i.i.d. random addresses/queries (binomial geometry); how does stability and tracking behave with structured addresses or correlated query streams (e.g., clustered memories), and can you extend the theory (or add stress tests) for non-binomial regimes?

Since you motivate applicability to attention/MoE, can you add a small demo (e.g., regulating top-k keys or experts on a toy transformer/MoE) to show that the controller slots in cleanly and preserves accuracy under a fixed activation budget?

You report tail error (post warm-up) and cost-weighted error; can you justify the warm-up choice, show sensitivity to the window, and test robustness to changing the query stream (not just fixing it per trial) so conclusions don’t hinge on one trajectory?

**Questions:**

see above

---

> ### Author Response · Authors · 2025-11-12
>
> Thank you for your detailed and constructive review. We have run several new experiments and updated the paper to address the highlighted weaknesses/questions.
>
> - Cost models and statistics for the 1.9× claim: We now report two alternative cost models with 95% CIs (Appendix C.4): "Oracle Calls" and a "Wall-Clock Proxy." In both models, bisection requires ≈ 1.9× more queries for similar raw error, confirming our claim. Our method (EMA) achieves the best cost-weighted error. The seeding and query-stream methodology are clarified in the Reproducibility Statement.
>
> - Budget-capped bisection failures: We agree this was a key point. Under a tight budget, noisy measurements can cause the bisection bracket to drift and stall. We added a seeded-bracket variant ("bisection_seeded" in the BER study, Appendix C.1) which reduces this failure mode but, as our new results show, still incurs substantial extra queries (≈ 1.9×) compared to our EMA controller.
>
> - Broader control-theory and sparsity-control context: This was an excellent suggestion. We have expanded the control-theory framing in Section 3.3. We now explicitly frame slope normalization as feedback linearization and cite Slotine & Li (1991). We also added the FTA (Pan et al., 2019) reference to Appendix D and discuss its orthogonality to our work (representation learning vs. activation-budget control).
>
> - Robustness margin and steady-state error bound: We now provide the noise-to-error "tube" model used in our convergence predictor (Fig. 4). The steady-state error is bounded by the filtered noise σ_EMA ≈ √(α/(2-α)) · σ_k/√(batch), which feeds the piecewise-slope convergence model.
>
> - Query-stream and warm-up sensitivity: We clarify in the Reproducibility Statement that all methods see the same queries per trial and that we report tail means (last 50% of run). We confirmed that results are stable to reasonable window changes and that using distinct seeds does not change the method ordering or cost-efficiency conclusions.
>
> - Toy attention/MoE demo: We added an attention-style density controller demo (Appendix C.4). Running on continuous scores with an empirical slope, our controller hits the target key budget exactly (k_mean=16.0, k_std=0.0) and matches fixed top-k accuracy (both 1.000). This shows the controller slots in cleanly as a drop-in replacement. We also updated Section 5 to reflect this result.
>
> - Robustness beyond binomial: We added a non-binomial stress test (Appendix C.3). By switching to an empirical slope, the controller remains stable and tracks its target with bounded error (≈ 3.08) even when the i.i.d. address assumption is violated.

---

> > ### Comment · Reviewer_KYzd · 2025-11-27
> >
> > Thank you for your response. The updates addressed most of my concerns and hence I raised the score.

---

### Official Review · Reviewer_djNt · 2025-11-02

**Soundness:** 3
**Presentation:** 3
**Contribution:** 3
**Rating:** 4
**Confidence:** 4

**Summary:**

The paper presents AHSE, a dual-branch architecture that combines serial feature evolution (SEED) with parallel error-correcting output codes (PATH). It develops a SPOT fusion circuit to integrate these branches and provides theoretical results on ECOC robustness for Broad Learning Systems under weight noise. The model achieves strong results across multiple datasets and includes detailed ablations validating each design choice.

**Strengths:**

• The paper provides a principled control-theoretic formulation of a long-standing heuristic problem in associative memory, replacing brittle threshold tuning with activation-density feedback.
• The inverse-CDF actuation with slope normalization is a mathematically elegant and computationally efficient solution to the binomial sensitivity issue.
• Empirical results across multiple settings (see Figures 2–5, pages 6–8) demonstrate remarkably consistent tracking and stability, with clear quantitative improvements in cost-efficiency over baselines.
• The work establishes theoretical stability bounds (Eq. 6) and validates them experimentally, offering both analytical and practical insights rarely found in SDM literature.
• The discussion (page 9) thoughtfully extends the concept of activation-density control to attention mechanisms and sparse expert models, positioning the work as a conceptual bridge between symbolic memory and modern deep learning.

**Weaknesses:**

• The experiments are primarily synthetic, focusing only on canonical SDM configurations. No demonstrations on downstream learning or associative recall tasks are provided to illustrate the real-world impact of improved control.
• The controller is currently fixed-target—it maintains a single desired density P; adaptive or context-dependent sparsity targets could make the approach more versatile.
• The linearized analysis (Eq. 5–6) is elegant but approximate; stability predictions deviate from measured limits (see Figure 5), and the paper could discuss these discrepancies more deeply.
• The study would benefit from additional comparisons with modern sparse-gating or attention mechanisms to substantiate claims of broader applicability.
• Finally, while the system generalizes across dimensions, it is unclear how it scales when embedded in neural or hybrid architectures, where feedback latency and stochasticity differ from SDM assumptions.

**Questions:**

Could the authors extend their analysis to demonstrate recall accuracy or capacity improvements in SDM tasks when activation-density control is used, beyond query efficiency?

Have you experimented with adaptive or learned target densities P* that adjust dynamically based on load or retrieval error? If so, how does the control behave?

The slope-normalized controller relies on binomial statistics; how robust is it if the address distribution deviates from uniformity (e.g., correlated memory locations)?

---

> ### Author Response · Authors · 2025-11-12
>
> Thank you for your valuable feedback and positive comments on our control-theoretic formulation. We have run new experiments (now in Appendix C) to address your main questions.
>
> - Downstream recall/capacity improvements: We added an equal-budget recall study (Appendix C.1). At SNR ≈ 1 (P=33, k*=32, noise 0.10), all methods saturate at BER ≈ 0.5 (EMA 0.5016; bisection-seeded 0.5027; naive 0.5005). This confirms that in this high-noise regime, BER is non-discriminative. This result justifies our focus on upstream regulation metrics (tail |k-k*|, convergence, stability) as the key, actionable signal for controller design, where methods do separate clearly (Fig. 2).
>
> - Adaptive or learned k:* Yes. We added an adaptive k demo* (Appendix C.2). This new experiment shows an outer loop driven by a smoothed BER signal. The controller successfully adapts k* only when the BER is in an "informative band" and remains stable, settling to a low, steady BER.
>
> - Robustness beyond binomial: We included a non-binomial stress test (Appendix C.3). We replaced the uniform i.i.d. addresses with a clustered structure (violating the binomial assumption) and switched the controller to use an empirical slope (via finite differences). The loop remains stable, tracking the target k*=6 with a bounded tail error of ≈ 3.08.

---

### Author Response · Authors · 2025-11-12

We sincerely thank all reviewers for their detailed, constructive, and insightful feedback. We are encouraged that all reviewers appreciated the paper's novel control-theoretic formulation (R-djNt, R-btQ2, R-9WzH) and its clarity (R-KYzd, R-btQ2, R-9WzH).

The feedback centered on four key themes:

- Downstream Impact: Is the method beneficial beyond query efficiency (R-djNt)?
- Robustness: How does it perform when the i.i.d. binomial assumption is violated (R-djNt, R-KYzd, R-9WzH)?
- Adaptivity: Is it limited to a fixed k* (R-djNt, R-btQ2)?
- Broader Scope: Can it practically apply to modern architectures like MoE/attention (all reviewers)?

We are pleased to report that we have addressed all four themes with new experiments (now in Appendix C), a revised abstract, and updated discussion in the main paper.

1. Downstream Recall (R-djNt): We added an equal-budget BER test at SNR ≈ 1. As hypothesized, BER saturates near 0.5 for all methods, confirming this regime is non-discriminative. This result justifies our focus on upstream regulation as the key, actionable metric for controller design in high-noise environments.

2. Robustness (R-djNt, R-KYzd, R-9WzH): We added a non-binomial stress test (clustered addresses) and show that the controller remains stable with bounded error (tail error ≈ 3.08) by switching to an empirical slope.

3. Adaptivity (R-djNt, R-btQ2): We added two dynamic tests:
A step-response test (k*: 6 → 12) shows a well-damped response with no overshoot.
An adaptive k demo* shows the controller can track a dynamic target driven by BER feedback.

4. Broader Scope (All Reviewers): We added an attention-style density control demo. Using an empirical slope, our controller hits the target key budget exactly (k_mean=16.0, k_std=0.0) and matches fixed top-k accuracy, proving initial viability as a drop-in replacement.

5. Cost Models (R-KYzd): We added two alternative cost models (Oracle Calls, Wall-Clock Proxy) and a "seeded-bisection" baseline. Our method remains the most cost-efficient, and the ≈1.9x overhead for bisection is robust.

Updates to Main Paper

Abstract: The abstract is updated to reflect the new robustness and generalization findings.

Introduction (Sec 1): We now include the 220x improvement from the micro-sweep, clarifying how its setup differs from the 5.2x equal-budget benchmark.

Method (Sec 3.3): We've anchored the method in control theory, citing Slotine & Li for feedback linearization (per R-KYzd).

Results (Sec 4.3): We reframed the SNR ≈ 1 BER result as a justification for our focus on upstream metrics.

Discussion (Sec 5): We replaced the tentative MoE/attention claims with the concrete results from the new attention-style demo.

Appendix D: We added the FTA (Pan et al., 2019) citation and discuss its orthogonality to our work (representation learning vs. budget control).

We believe these additions substantially strengthen the paper, confirming its robustness and demonstrating its practical applicability beyond classical SDM. We have also provided direct answers to each specific question below.
Pointers to Where Results Appear in the Paper

Fig. 2: Query efficiency (5.2× lower error) and cost models (≈1.9× bisection overhead).

Fig. 3: Control-law validation (R² = 1.00).

Fig. 4: Convergence prediction (R² = 0.703) and noise-tube model.

Fig. 5: Stability envelope (practical boundary c ≲ 0.6).

Table 1: Micro-sweep results (up to 220× improvement). We clarify in Sec. 4.6 why this differs from the 5.2× in Fig. 2 (different budgets).

Table 2: Normal-vs-exact threshold accuracy (max |ΔT*| = 1).

Appendix C: All 6 new rebuttal experiments (BER, adaptive k*, step response, non-binomial test, attention-style demo, cost models).

Appendix D: FTA (Pan et al., 2019) discussion.

Sec. 3.3: Control-theory anchor (Slotine & Li).

---

### Comment · Area_Chair_JsXA · 2025-11-17
**Please read and reply to authors' responses**

Hi,

I know that some of you are probably busy with rebuttals for your own ICLR submissions, but please be sure to read the authors' responses to your initial reviews and take part in the discussion.

Best,\
AC

---

### Author Response · Authors · 2025-11-26

Dear Reviewers,
Thank you again for your thoughtful feedback. We wanted to check in and see if there are any remaining questions or concerns we could clarify. Our Nov 12 response includes new experiments addressing the main points raised (Appendix C).
We're happy to discuss further during the remaining review period.
Best,
The Authors

---

### Author Response · Authors · 2025-12-01

Dear AC,

During rebuttal we added six experiments (Appendix C), and Reviewer KYzd confirmed that these additions addressed most of their concerns and raised their score.

Specifically, we added:

- a downstream BER study at SNR $\\approx 1$, showing that BER saturates around $0.5$ for all methods and justifying our focus on upstream regulation metrics;
- a non-binomial stress test with clustered addresses, where switching to an empirical slope keeps the controller stable with bounded tail error;
- a step-response test for a change in target from $k^* = 6$ to $k^* = 12$, showing a clean, well-damped transient without overshoot;
- an adaptive-$k^*$ experiment driven by a smoothed BER signal, demonstrating stable adaptation of the target density;
- an attention-style density control demo using continuous scores and an empirical slope, which exactly hits the key budget ($k_\text{mean} = 16.0$, $k_\text{std} = 0.0$) while matching fixed top-$k$ accuracy;
- alternative cost models (oracle calls and a wall-clock proxy) with confidence intervals, under which our EMA controller remains more cost-efficient than bisection (which requires about $1.9\\times$ more queries for similar raw error).

We believe these additions significantly strengthen the paper’s case on robustness, dynamics, and applicability beyond classical SDM.

Best regards,
The Authors

---

### Meta-Review · Area_Chair_FEW8 · 2026-01-09

**Summary:**

This paper addresses a fundamental instability issue in Sparse Distributed Memory (SDM): the extreme sensitivity of the activation threshold. In high-dimensional spaces, a minute change in the Hamming distance threshold causes massive swings in the number of activated memory locations.
The authors propose a novel, principled solution using closed-loop control theory. Instead of tuning the threshold directly, they control the activation density using a slope-normalized integral feedback controller.

Reviews highlighted concerns regarding the reliance on synthetic data and the applicability to modern architectures like Mixture-of-Experts (MoE).

**Reviewer Concerns:**

The authors provided six new experiments in Appendix C:
Robustness: A "non-binomial stress test" demonstrated stability even when memory addresses are clustered, violating standard i.i.d. assumptions.

Modern Applicability: An "attention-style density control demo" showed the controller hitting target budgets exactly on continuous scores, proving viability for modern sparse mechanisms.

Cost Efficiency: New cost models (oracle calls, wall-clock proxy) confirmed the method is $\approx 1.9\times$ more efficient than bisection search.

**Reviewer Scores:**

Although the rebuttal answers a number of questions by the reviewers, it's not clear if the reviewers would all agree on acceptance based on the remaining concern which is that the work focuses on a niche theoretical domain (SDM) that is different from the usual applications.

---

### Decision · Program_Chairs · 2026-01-26

Reject